# Safety of Anti-Reelin Therapeutic Approaches for Chronic Inflammatory Diseases

**DOI:** 10.3390/cells13070583

**Published:** 2024-03-27

**Authors:** Laurent Calvier, Anna Alexander, Austin T. Marckx, Maria Z. Kounnas, Murat Durakoglugil, Joachim Herz

**Affiliations:** 1Department of Molecular Genetics, University of Texas (UT) Southwestern Medical Center, Dallas, TX 75390, USAaustin.marckx@utsouthwestern.edu (A.T.M.); murat.durakoglugil@utsouthwestern.edu (M.D.); joachim.herz@utsouthwestern.edu (J.H.); 2Center for Translational Neurodegeneration Research, UT Southwestern Medical Center, Dallas, TX 75390, USA; 3Reelin Therapeutics Inc., La Jolla, CA 92130, USA; maria.kounnas@reelintherapeutics.com; 4Department of Neuroscience, UT Southwestern Medical Center, Dallas, TX 75390, USA; 5Department of Neurology and Neurotherapeutics, UT Southwestern Medical Center, Dallas, TX 75390, USA

**Keywords:** Reelin, ApoER2, NF-κB, vascular system, immune system, fibrosis, inflammation, leukocyte, multiple sclerosis, atherosclerosis

## Abstract

Reelin, a large extracellular glycoprotein, plays critical roles in neuronal development and synaptic plasticity in the central nervous system (CNS). Recent studies have revealed non-neuronal functions of plasma Reelin in inflammation by promoting endothelial–leukocyte adhesion through its canonical pathway in endothelial cells (via ApoER2 acting on NF-κB), as well as in vascular tone regulation and thrombosis. In this study, we have investigated the safety and efficacy of selectively depleting plasma Reelin as a potential therapeutic strategy for chronic inflammatory diseases. We found that Reelin expression remains stable throughout adulthood and that peripheral anti-Reelin antibody treatment with CR-50 efficiently depletes plasma Reelin without affecting its levels or functionality within the CNS. Notably, this approach preserves essential neuronal functions and synaptic plasticity. Furthermore, in mice induced with experimental autoimmune encephalomyelitis (EAE), selective modulation of endothelial responses by anti-Reelin antibodies reduces pathological leukocyte infiltration without completely abolishing diapedesis. Finally, long-term Reelin depletion under metabolic stress induced by a Western diet did not negatively impact the heart, kidney, or liver, suggesting a favorable safety profile. These findings underscore the promising role of peripheral anti-Reelin therapeutic strategies for autoimmune diseases and conditions where endothelial function is compromised, offering a novel approach that may avoid the immunosuppressive side effects associated with conventional anti-inflammatory therapies.

## 1. Introduction

Reelin is a large extracellular glycoprotein initially known as a neuron guidance protein critical for neuronal development and migration, especially in the cerebellum, cortex, and hippocampus [1,2,3,4,5,6,7,8,9,10]. In the prenatal brain, Reelin is mainly secreted by Cajal–Retzius cells throughout embryonic development where it regulates neuroblast migration and the growth of layer-specific connections in the hippocampus and entorhinal cortex [11,12,13,14,15,16,17,18,19,20]. In the adult brain, it is mainly secreted by a subset of cortical GABA-ergic interneurons, cerebellar granule cells, and hippocampal interneurons [21,22] and modulates synaptic plasticity [14,23], migration of neuroblasts [24], as well as dendrite [25] and dendritic spine [26] formation. Reelin expression in the central nervous system (CNS) has been implicated in a wide range of neurodegenerative diseases, such as Alzheimer’s disease (AD), cognitive disorders, such as autism spectrum disorder, and neuropsychiatric disorders, such as schizophrenia and bipolar disorder [15,27,28,29,30,31,32,33,34,35,36,37,38,39]. The nascent Reelin protein consists of a signal peptide, succeeded by an F-spondin-like domain, eight Reelin repeats featuring an epidermal growth factor (EGF)-like cysteine pattern that separates each repeat into subdomains A and B, and finally, a positively charged C terminus [40,41]. Physiologic proteolytic cleavage of Reelin [7,8,42,43,44,45,46,47,48,49,50] occurs between Reelin repeats 2 and 3, and another between repeats 6 and 7 [41,45,48,50,51,52], which separates interaction sites with various cell surface receptors [21,22,41], but much about the physiological significance of these cleavage events remains unknown.

Reelin is also abundant in the blood with a typical concentration of around 10 ng/mL in humans [53], and exercises function outside of the CNS [51,54,55,56,57,58,59,60,61,62,63,64,65,66,67,68,69,70,71]. In the past 10 years, three major non-neuronal functions for plasma Reelin have been described in inflammation, vascular tone regulation, and thrombosis. First, we have demonstrated in models of atherosclerosis [58,72], multiple sclerosis (MS) [53,73], and COVID-19 [74] that plasma Reelin inactivation reduces inflammatory cell recruitment by decreasing the expression of leukocyte–endothelial adhesion proteins (E-selectin, ICAM-1, and VCAM-1). These inflammatory mediators in endothelial cells are increased by Reelin through one of its receptors, the Apolipoprotein E receptor-2 (ApoER2 or LRP8) [73,75,76], and NF-κB [77,78,79,80,81,82,83,84,85,86,87,88,89,90,91,92,93], regulating endothelial homeostasis [58]. Secondly, Reelin antagonizes endothelial NOS activity, which regulates NO synthesis and vasodilation [58]. Thirdly, Reelin stimulates platelet adhesion by interacting with coagulation factors such as GPVI, promoting thrombosis [57,60,66,67,94]. The source of plasma Reelin is unclear, especially in diseases [41], but the hepatic stellate cells are believed to secrete a significant fraction of it [69,70,95,96,97]. These results suggest a role for this circulating protein in chronic inflammatory diseases with a compromised endothelium and highlight its potential as a therapeutic target [98].

Several models and tools have been developed to study and deplete Reelin. In 1951, a new strain of mice was discovered with a reeling gait and inside-out cortical layering later attributed to an autosomal recessive mutation. This mouse strain was accordingly given the name *reeler* [99,100,101,102,103,104,105,106,107,108]. In 1995, the RELN gene was identified on chromosome 7q22 and successfully cloned, leading to the discovery of its deletion in *reeler* mice [109]. The same year, a mouse monoclonal anti-Reelin antibody was developed [110], having the remarkable ability to block Reelin function [110,111,112,113,114,115,116], but also to deplete it from the circulation when administered by intravenous or intraperitoneal injection [53,72,74]. This antibody, named CR-50, was generated by immunizing reeler mice with brain homogenate from control embryonic mice [110]. It blocks Reelin signaling by binding to an epitope composed of amino acids 230–346 on the N-terminus of Reelin [110,114,115]. In 2015, Reelin floxed mice were developed by our laboratory [117], allowing the creation of conditional KO mice to circumvent the impaired neurological development inherent to the *reeler* mice. Finally, in 2021, anti-sense oligonucleotides (ASO) specific to mouse Reelin mRNA were designed by Ionis Pharmaceuticals and we demonstrated their efficacy in blocking Reelin protein expression systemically [72].

The potential of anti-Reelin therapeutic strategies for the treatment of chronic inflammatory diseases is the subject of ongoing research. Reelin depletion has been identified as a novel therapeutic approach that selectively targets the vascular adhesion of leukocytes; thus, we hypothesize that this target potentially offers a superior safety profile in conditions marked by chronic inflammation. In this study, we have tested the safety of plasma Reelin depletion to demonstrate the low risk of Reelin-depleting therapeutic strategies.

## 2. Materials and Methods

### 2.1. Human Cohort

The serum of anonymous healthy controls was obtained from UT Southwestern Medical Center MS tissue repository (authorization #STU022011-211, Dallas, TX, USA). The study presented here has been approved by the “Human Research Protection Program Office” as “Not Human Research”, which does not require IRB approval.

### 2.2. Experimental Autoimmune Encephalitis (EAE) Model

This study uses tissues collected previously [53]. Briefly, *Cx3cr1*-GFP mice (B6.129P-Cx3cr1^tm1Litt^/J) were purchased from the Jackson Laboratories (Stock No. 005582). These mice express EGFP in monocytes, dendritic cells, NK cells, and brain microglia under the control of the endogenous *Cx3cr1* locus. *Cx3cr1*-GFP monocytes downregulate GFP expression upon differentiation into macrophages.

Eight-week-old male mice were immunized by subcutaneous injection of myelin basic protein peptide MOG35-55 (200 µg; Sigma-Aldrich (St. Louis, MO, USA); M4939) in 200 µL of emulsified complete Freund’s adjuvant (Sigma-Aldrich; F5506) containing 2 mg/mL of *Mycobacterium tuberculosis* (Fischer Scientific (Waltham, MA, USA); DF3114-33-8). In addition, 400 ng of pertussis toxin (Fisher Scientific; NC9282261) was administered intraperitoneally on day 0 and day EAE severity was blindly evaluated daily from day 7 by survival, EAE clinical score (from 0 = healthy to 10 = dead), weight loss, and hanging test (inverted grid (8 × 8 mm) test for a maximum time of 180 s). On day 21, the mice were sacrificed as stipulated by the UT Southwestern Institutional Animal Care and Use Committee (IACUC), blood was collected for plasma analysis, the spinal cord was extracted by hydraulic pressure and cut into 3 pieces (sub-lumbar and sub-cervical parts for histology, the thoracic section for protein analysis).

### 2.3. Aging Mice under a Western Diet

Cx3cr1-GFP, Cag-Cre Reln^fl/lf^, and Ldlr^−/−^ lines were crossed to obtain the Cx3cr1-GFP; Cag-Cre Reln^fl/f^; Ldlr^−/−^ mice. Tamoxifen induction was started at 7 weeks old and at 8 weeks old male and female mice were placed on a Western diet (Envigo; ref TD.88137; with cholesterol [0.2% total cholesterol], total fat [21% by weight; 42% kcal from fat], high in saturated fatty acids [>60% of total fatty acids], high sucrose [34% by weight]). At 6 or 12 months of age, mice were euthanized by anesthetic overdose, hearts were perfused with saline solution, and flushed tissues were fixed in 4% PFA.

### 2.4. ELISA

Reelin ELISA was performed on human serum samples diluted at 1:30 and performed according to the manufacturer’s instructions (LSBio, N-Terminal part: LS-F7023).

### 2.5. Immunofluorescence

As previously published [53], tissues were fixed overnight in 4% paraformaldehyde and then cryoprotected in sucrose solutions (15, then 30% in PBS), followed by being embedded in OCT (Tissue-Tek (Torrance, CA, USA); 4583) and frozen at −80 °C. Then, 10 µm thick cryosections were cut using a Cryostat (Leica (Wetzlar, Germany)). Sections were thawed, rehydrated in PBS, blocked for 1 h at room temperature with blocking buffer (1% BSA and 0.03% Triton-X100 in PBS), incubated with primary antibody (ICAM-1, R&D Systems (Minneapolis, MN, USA), AF796; Iba1, Novus Biologicals (Littleton, CO, USA), NB100-1028) overnight at 4 °C in a humid chamber, washed with PBS, incubated with Alexa594-conjugated secondary antibody (anti-goat, Molecular Probes (Waltham, MA, USA), A11058) for 1 h at room temperature, washed, and mounted using Vectashield antifade mounting medium with Dapi (Vector, H-1200). Images were acquired with a Zeiss (Oberkochen, Germany) Axiophot microscope with AxioVision software (version 4.8) and analyzed with ImageJ (version 1.54).

### 2.6. Immunoblotting

As previously published [53], cell lysates or tissue pieces were prepared by adding protease and phosphatase inhibitor cocktail in RIPA buffer and centrifuging for 10 min at 12,000 rpm to remove debris. From the lysates, protein concentrations were determined by the Lowry protein assay (500-0113, 500-0114, 500-0115; Bio-Rad (Hercules, CA, USA)). Equal amounts of protein were loaded into each lane of a 4–12% Tris gel (BioRad) and subjected to electrophoresis. After blotting, nitrocellulose membranes (BioRad) were blocked for 1 h (milk powder 5% in TBS/tween 0.1–0.2%) and incubated with primary antibodies (ICAM-1, R&D Systems, AF796; E-Selectin, Santa Cruz, sc-137054; G10 anti-Reelin, made in house; Reelin, Millipore, MAB5366; GAPDH, Sigma-Aldrich, G8795). The binding of secondary HRP-antibodies was visualized by ECL or ECL plus chemiluminescent (Amersham). After densitometric analyses with ImageJ, optical density values were expressed as arbitrary units and normalized for protein loading, as described in the figure legends.

### 2.7. Field Electrophysiology and θ-Burst Long-Term Potentiation (LTP)

θ-Burst LTP was performed as previously described [117]. Briefly, mice were deeply anesthetized with isoflurane, the blood was flushed out with ice-cold saline solution, and brains were quickly removed and placed in ice-cold high-sucrose slicing solution (110 mM sucrose, 60 mM NaCl, 3 mM KCl, 1.25 mM NaH_2_PO_4_, 28 mM NaHCO_3_, 0.5 mM CaCl_2_, 5 mM glucose, 0.6 mM ascorbic acid, 7 mM MgSO_4_). Transverse slices (350 mm) were cut using a Leica VT1000S vibratome. Slices were allowed to recover in ACSF (124 mM NaCl, 3 mM KCl, 1.25 mM NaH_2_PO_4_, 26 mM NaHCO_3_, 10 mM D-glucose, 2 mM CaCl_2_, 1 mM MgSO_4_) for 1 h at room temperature before the experiments. For recording, slices were transferred to an interface chamber perfused with ACSF at 31 °C. Slices were stimulated in the stratum radiatum with concentric bipolar electrodes (FHC) using an isolated pulse stimulator. The stimulus intensity was set to 40 to 60% of the maximum peak amplitude, as determined by measuring the input–output curve. After the baseline stabilized, a theta burst (TBS) was applied using a train of four 100 Hz pulses repeated 10 times with 200 ms intervals and the train was repeated five times at 10 s intervals. The resulting LTP was measured for 1 h after theta-burst stimulation. Data were analyzed using LabView 7.0.

### 2.8. Statistical Analysis

N numbers are specified in each legend. GraphPad Prism software 9.4.0 was used to run all the statistical analyses. Values from multiple experiments are expressed as means ± SEM. Normality was tested using the Kolmogorov–Smirnov test. Statistical significance was determined for multiple comparisons using one-way analysis of variance (ANOVA), followed by Tukey’s multiple comparisons (for normal distribution) or Kruskal–Wallis (for non-normal distribution) test. Student’s *t*-test (for normal distribution) or Mann–Whitney (for non-normal distribution) were used for comparisons of the two groups. The correlations were calculated by linear regression (Pearson’s r). *p* values lower than 0.05 were considered significant, with * *p* < 0.05, ** *p* < 0.01, *p* < 0.001, and lower not marked specifically and included in ** *p* < 0.01.

## 3. Results

### 3.1. Reelin Is Stably Expressed throughout Adulthood

Reelin is found abundantly present in the bloodstream, typically at a concentration of approximately 10 ng/mL in healthy humans [53]. On average, Reelin levels rise two to threefold under chronic or low-grade inflammatory conditions [53,74], and up to fivefold during acute or severe inflammation, as observed in COVID-19 patients experiencing a cytokine storm [74]. With the increasing prevalence of chronic inflammatory diseases with age, we wanted to test whether baseline Reelin expression changes over time. ELISA and Western blot analysis, in human and mouse cohorts, respectively, revealed that Reelin expression in the circulation remains stable throughout adulthood (Figure 1A,B). This suggests that the increased secretion of this plasma protein during inflammatory conditions is not related to aging but directly to the inflammatory processes.

### 3.2. Peripheral Reelin Depletion Preserves Its Expression and Function in the CNS

Preserving Reelin expression in the brain is important, as Reelin is not only essential for neuronal migration and positioning during development [41] but also in adults where it modulates synaptic plasticity [14,23], migration of neuroblasts [24], as well as dendrite [25] and dendritic spine [26] formation. We have shown that Reelin, a large glycoprotein (~400 kDa), does not cross the BBB nor translocate between the CNS and the circulation [58]. We have previously shown that we can deplete Reelin from plasma by injecting Reelin^fl/fl^ mice through the tail vein with an adenovirus expressing Cre recombinase (Ad-Cre) [58]. Immunoblot analysis demonstrated efficient and specific ablation of this protein from plasma, but not from the brain. Nevertheless, preserving Reelin expression in the CNS is important, where Reelin has crucial functions in the regulation of synaptic neurotransmission and network homeostasis [117]. This necessitated a thorough investigation of whether peripheral Reelin depletion affects Reelin levels and function in the CNS in any significant way.

First, to test if the anti-Reelin antibody CR-50 interferes with CNS Reelin, we have challenged the BBB and increased its permeability using the experimental autoimmune encephalomyelitis (EAE) mouse model. Bi-weekly CR-50 intraperitoneal injections for three weeks efficiently deplete Reelin from the plasma, but not from the brain or the spinal cord (Figure 2A–C).

Then, beyond CNS Reelin depletion, we tested if CR-50 intraperitoneal injections could interfere with Reelin function in the brain. We have previously shown that after the end of the neurodevelopmental period, Reelin knockdown in a conditional knockout mouse model causes abnormal elevations in theta-burst-induced long-term potentiation (LTP) in the hippocampal CA1 region [117], a special form of synaptic plasticity used as a model for learning and memory. Here, we used the same protocol to induce LTP in mice treated with CR-50 i.p. injections to deplete circulating, but not intracerebral, Reelin (Figure 2D). We did not detect any difference in theta-burst-induced LTP between brain slices from CR-50 (1.396 ± 0.20, *n* = 11) and control IgG-treated mice (1.390 ± 0.11, *n* = 14). D-AP5 is an NMDA receptor blocker that can induce synaptic scaling [119,120]. Here, we showed that a blockage of NMDA receptors at rest disinhibits AMPA receptor insertion into the synapse and causes a rapid synaptic potentiation, which stabilizes within 25–35 min. We did not find any significant difference in D-AP5-induced scaling between CR-50 (1.74 ± 0.21, *n* = 12) and control IgG (1.49 ± 0.15, *n* = 7) brain slices (Figure 2E). Input–output curves were calculated using increasing stimulus intensities and normalized using presynaptic fiber volleys. A line was fitted using various points at the curve, which failed to show any significant differences (Figure 2F). Of note, no body weight difference was found following Reelin depletion by CR-50, or in conditional KO mice.

Hence, the peripheral administration of anti-Reelin antibodies effectively removes this protein from the bloodstream while preserving its levels and functionality within the central nervous system (CNS). This indicates that peripheral Reelin depletion is safe and does not affect its function in the CNS.

### 3.3. Plasma Reelin Depletion Restores Endothelial Function without Abolishing Diapedesis

Under normal conditions, the endothelial wall effectively regulates the passage of immune cells into the CNS. However, under inflammatory conditions such as MS, this barrier is compromised, becoming more adhesive and permeable, which dramatically increases leukocyte rolling, adhesion, and infiltration [121,122,123]. We have previously shown that plasma Reelin inactivation reduces inflammatory cell recruitment by decreasing the expression of leukocyte–endothelial adhesion markers (E-selectin, ICAM-1, and VCAM-1) [53].

Here, we show that peripheral CR-50 treatment in mice with EAE decreases the expression of adhesion proteins (E-selectin and ICAM-1) to a level no lower than that observed in littermate naïve mice (Figure 3A–C). Consequently, we observed in EAE mice that the pathological infiltration of leukocytes was reduced by CR-50 treatment to the level observed in naïve mice (Figure 3D). Since both resident microglia and infiltrating monocytes express *Cx3cr1*-GFP, the microglia-specific marker Iba1 was used to discriminate between the two populations and isolate the infiltrating monocytes.

Therefore, the peripheral administration of anti-Reelin antibodies effectively blocks excessive infiltration of leukocytes during inflammation while preserving a basal expression of endothelial–leukocyte adhesion proteins and diapedesis.

### 3.4. Plasma Reelin Depletion Does Not Adversely Affect Organ Function Outside the CNS

Previously, we showed that reduction of circulating Reelin, whether achieved through either antisense oligonucleotide (ASO) treatment or the use of the neutralizing antibody CR-50 over 16 weeks, significantly mitigated the development and progression of atherosclerotic plaques in *Ldlr*^−/−^ mice subjected to a high-cholesterol diet [72]. Building upon these findings, we have now expanded [72] our investigation to assess the effects of medium and long-term Reelin depletion on the heart, kidney, and liver in mice exposed to a Western diet. This model serves as a representation of prolonged chronic Reelin depletion in humans with a diet rich in cholesterol and calories.

To this end, Reelin conditional KO mice and WT littermates, both on an Ldlr KO background, were fed a Western diet for 4 months (for the 6-month time point) or 10 months (for the 12-month time point). At the end of the feeding periods, the heart, kidney, and liver were collected and stained with H&E and Masson’s trichrome (Figure 4). In the heart, no morphometric differences were found between the WT and Reelin KO groups within each time point, as judged by the thickness of the right ventricle, left ventricle, and septum (Appendix A). No difference was found for cardiomyocyte hypertrophy (transverse section area), interstitial and perivascular inflammatory infiltrates, and fibrosis. In the kidney, no difference was observed in the size of glomeruli, usually enlarged during kidney failure [124]. Finally, in the liver, although a clear fatty liver phenotype was observed for all groups, no differences were noted between WT and Reelin KO, as judged by interstitial and perivascular inflammatory infiltrates and fibrosis. Of note, no increased mortality or cancer occurrence was reported in the Reelin KO mice compared to WT.

These results suggest that long-term Reelin depletion under metabolic stress induced by a Western diet does not negatively impact the function of various organs, such as the heart, kidney, or liver.

## 4. Discussion

The objective of this study was to test the safety of plasma Reelin depletion as a potential therapeutic strategy for chronic inflammatory diseases. This study aimed to de-risk future anti-Reelin therapeutic strategies by investigating whether Reelin depletion would have adverse effects on organ functions and neuronal plasticity. To this end, we have shown that plasma Reelin expression is stable throughout adulthood in both humans and mice, suggesting that variations in its concentration during inflammatory conditions are more associated with ongoing inflammation than aging. Notably, peripheral anti-Reelin antibody treatment efficiently depleted plasma Reelin without affecting its levels or functionality within the CNS. This treatment approach demonstrated safety, as it did not interfere with essential CNS functions, such as synaptic plasticity. Moreover, the selective modulation of endothelial responses by anti-Reelin antibodies was evident, with a reduction in pathological leukocyte infiltration down to the control level. Finally, long-term Reelin depletion under metabolic stress induced by a Western diet did not negatively impact the heart, kidney, or liver, suggesting a favorable safety profile for potential therapeutic strategies targeting Reelin in chronic inflammatory diseases. Overall, the findings demonstrate the promising and safe potential of peripheral Reelin-depleting therapeutic strategies for inflammatory conditions with compromised endothelial integrity.

One pivotal aspect of our investigation involved the preservation of Reelin levels and functionality within the central nervous system (CNS) during peripheral anti-Reelin antibody treatment, as this protein participates in proper neuronal function [14,23,24,25,26,41]. Our findings demonstrate that, through targeted peripheral anti-Reelin antibody injections, efficient depletion of plasma Reelin can be achieved without compromising the brain’s Reelin levels and its crucial neuronal functions, as opposed to a whole-body Reelin knockout [117]. This underscores the feasibility and safety of utilizing anti-Reelin antibodies as a peripheral treatment strategy without compromising CNS functions. Outside of the CNS, the source for circulating Reelin is unclear, but it is suspected that it is mainly produced by hepatic stellate cells and released into the bloodstream [95], where this protein is found in significant amounts [53]. Therefore, plasma Reelin expression is used to evaluate the efficacy of anti-Reelin intervention.

In the treatment of chronic inflammatory diseases like MS, anti-inflammatory drugs play a pivotal role by targeting the underlying immune dysregulation. However, it is important to note that, while these drugs can effectively manage symptoms, their immunosuppressive nature may pose risks of increased susceptibility to infections and other adverse effects. For example, Natalizumab binds to α4-integrin on leukocytes [125,126], thereby abolishing their ability to infiltrate into the CNS, but this is associated with a high risk for often deadly progressive multifocal leukoencephalopathy caused by JC virus reactivation due to a critical loss of immune surveillance of the brain [125,126]. To stop inflammation, stepping away from immunotherapies may seem counterintuitive at first, but also offers distinct advantages. During chronic inflammatory diseases, the vascular barrier plays a crucial role as it becomes more adhesive and permeable to leukocytes [127,128]. Based on this observation, we propose that circulating Reelin should be exploited to restore the endothelium to its physiological state and, consequently, normalize but not abolish the passage of immune cells. In almost a decade of work with Reelin conditional KO mice and Reelin-depleting agents, we have not observed any increased susceptibility of Reelin-depleted mice to inflammatory diseases or infection. On the contrary, we have recently demonstrated that CR-50 treatment does not worsen but actually protects from severe COVID-19 outcomes following infection with SARS-CoV-2 [74]. Moreover, patients with RELN mutations, leading to low or loss of Reelin expression, show lissencephaly, abnormal neuromuscular connectivity, and congenital lymphoedema that are attributed to Reelin’s absence during development [129]. Importantly, no immunodepression has been reported in these patients, confirming murine data and indicating safety in targeting plasma Reelin in adults.

The targeting of the endothelial factor Reelin offers a multitude of advantages over conventional anti-inflammatory therapies. First, Reelin inhibition normalizes leukocyte infiltration without complete abolition. Moreover, Reelin depletion does not exhibit any obvious immunosuppressive side effects. Finally, the regulatory role of Reelin extends beyond endothelial adhesion/permeability to leukocytes [53,58,72,73,74,98], encompassing thrombosis [57,60,66,67] and vascular tone [58]. These are common risk factors for cardiovascular disease [130,131,132,133,134,135], pulmonary hypertension [136,137,138,139,140,141,142,143], and stroke [130], of which Reelin lowering could predictably affect in a favorable manner. However, the literature reveals a multifaceted role for Reelin in cancer development [41], with studies suggesting both promotional [144,145,146,147,148,149,150,151,152,153] and inhibitory [62,154,155,156,157,158,159] effects on malignant cell behavior depending on the cell type and affected organ. Therefore, dedicated studies should assess whether cancer might represent a medical contraindication for future anti-Reelin therapies.

## 5. Conclusions

In summary, our study shows that depleting plasma Reelin through peripheral antibody treatment is safe and potentially effective for chronic inflammatory diseases. Reelin levels remain stable in adulthood, and antibody treatment preserves its crucial functions in the central nervous system without adverse effects on synaptic plasticity. Moreover, Reelin depletion selectively modulates endothelial responses, by reducing pathological leukocyte infiltration while maintaining basal diapedesis. Long-term depletion under metabolic stress has not revealed any adverse effects on the physiology of vital organs. The translational relevance of this study is highlighted by the potential therapeutic implications of targeting Reelin in chronic inflammatory conditions. The ability to selectively modulate Reelin levels in the periphery opens avenues for exploring its therapeutic potential in diseases characterized by compromised endothelial function, such as atherosclerosis, multiple sclerosis, and even severe inflammatory conditions like COVID-19.

## Figures and Tables

**Figure 1 cells-13-00583-f001:**
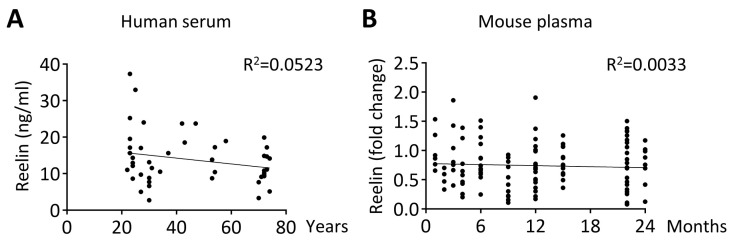
**Reelin expression is stable between sex and during aging.** (**A**) Reelin concentration was evaluated by ELISA in healthy patients (sex matched); *n* = 46 with 21 women and 25 men. (**B**) Reelin expression was evaluated by Western blot in WT mice at different ages.

**Figure 2 cells-13-00583-f002:**
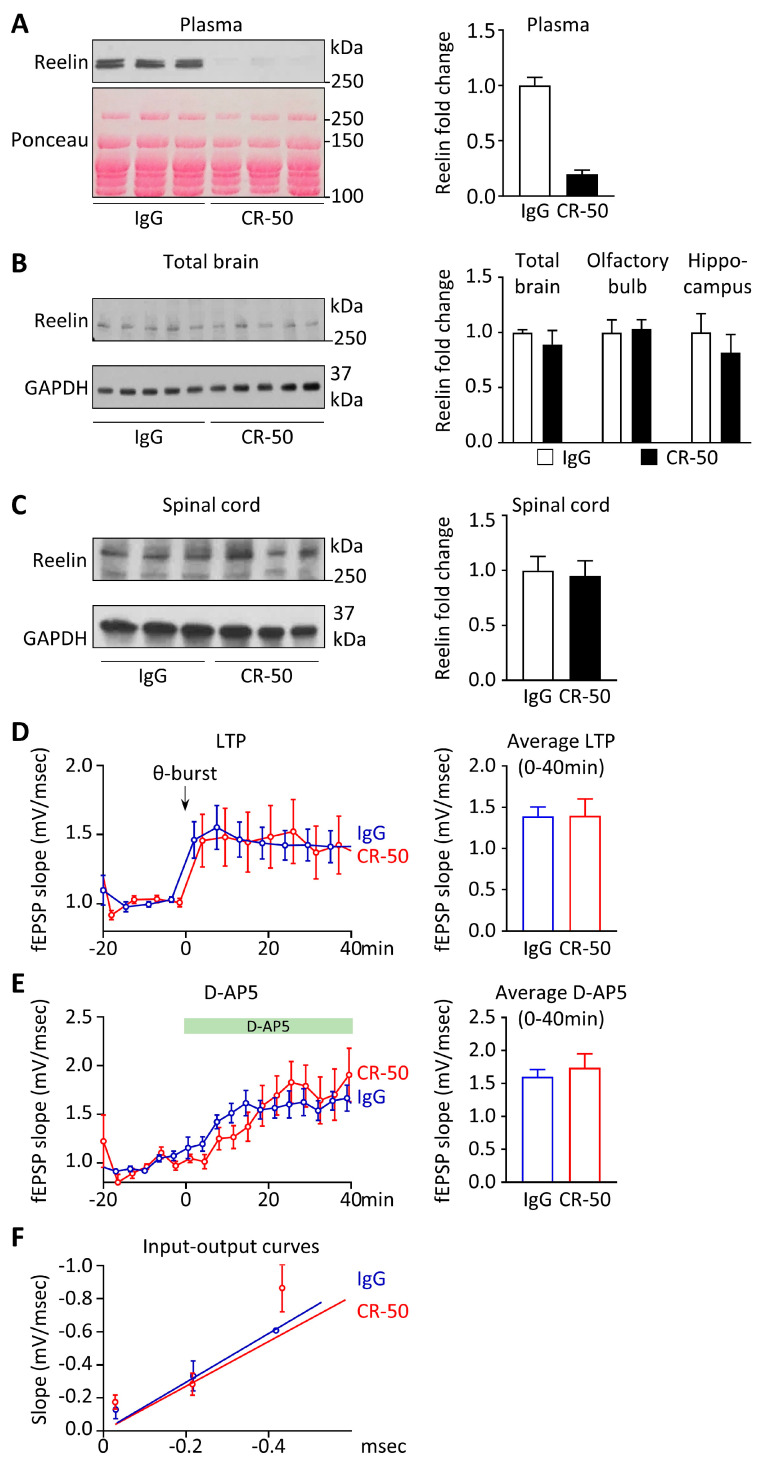
**Peripheral CR-50 treatment does not affect Reelin expression and function in the CNS.** (**A**–**C**) *Cx3cr1*-GFP male mice were injected intraperitoneally with 100 µg of irrelevant IgG (*n* = 10) or CR-50 (*n* = 9) twice per week. To challenge the BBB and increase its permeability, EAE was induced by Myelin Oligodendrocyte Glycoprotein immunization using standard procedures [118], one week after the first antibody injection. (**A**) Reelin protein expression was evaluated in plasma by Western blot (*n* = 9–10). (**B**) One total brain hemisphere was lysed and Reelin protein expression was evaluated by immunoblotting (*n* = 5). Reelin expression was also evaluated by immunoblotting in the olfactory bulb and the hippocampus (*n* = 3). (**C**) Reelin protein expression in the spinal cord was evaluated by Western blot in both groups. (**A**–**C**) Adapted with permission from Calvier et al. Sci. Trans. Med., 2020 [53]. (**D**–**F**) WT mice were injected intraperitoneally with 100 µg of irrelevant IgG (*n* = 10) or CR-50 (*n* = 9). (**D**) LTP induced by theta-burst stimulation (TBS, 200 pulses in total) in the CA1 region of hippocampal slices from CR50 and control mono IgG-treated mice. Baselines were normalized to 1 by dividing each experiment by its 10 min average before the TBS paradigm. LTP was calculated as the average potentiation between 40 and 60 min after TBS. We did not find any significant difference between CR50 (1.396 ± 0.20, *n* = 11) and IgG-treated slices (1.390 ± 0.11, *n* = 14). (**E**) DAP5 is an NMDA receptor inhibitor. Block of NMDA receptors at rest disinhibits AMPA receptor insertion into the synapse and causes a rapid synaptic potentiation which stabilizes within 25–35 min. After a 20 min stable baseline, AP5 was applied to induce rapid synaptic scaling. There were no significant differences between CR50 (1.74 ± 0.21, *n* = 12) and IgG (1.49 ± 0.15, *n* = 7)-treated slices. (**F**) Input output curves are plotted as increasing stimulus intensities, which are normalized to fiber volley amplitudes vs. fEPSP slopes and best fitted with a line which showed no significant differences at various points.

**Figure 3 cells-13-00583-f003:**
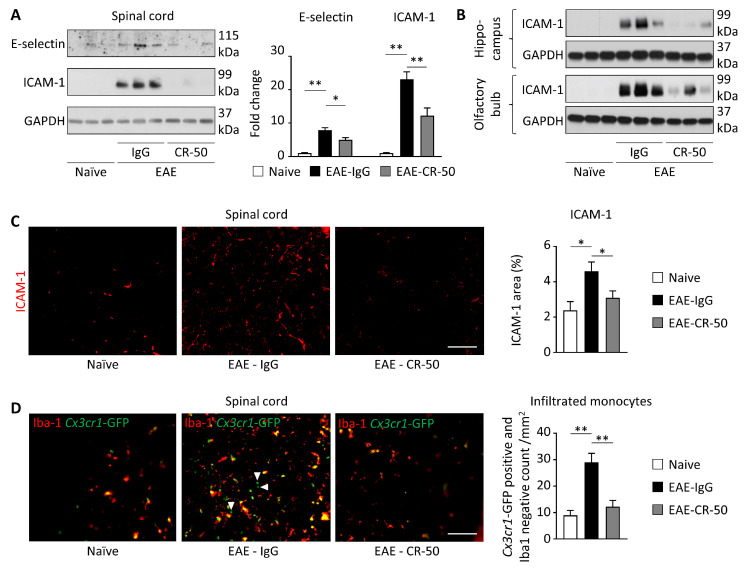
**Peripheral CR-50 treatment does not abolish basal diapedesis.** *Cx3cr1*-GFP male mice were injected intraperitoneally with 100 µg of irrelevant IgG (*n* = 10) or CR-50 (*n* = 9) twice per week. To challenge the BBB and increase its permeability, EAE was induced by Myelin Oligodendrocyte Glycoprotein immunization using standard procedures [118], one week after the first antibody injection. Naïve mice are littermates with no antibody treatment or EAE induction. (**A**–**C**) Adhesion protein expression was evaluated by Western blot or immunohistochemistry in tissues, as indicated. (**D**) In the *Cx3cr1*-GFP-positive cell population, the total number of inflammatory cells (*Cx3cr1*-GFP-positive), monocytes (*Cx3cr1*-GFP-positive, Iba-1-negative; indicated by the arrows), and microglia (*Cx3cr1*-GFP and Iba1 double positive) were visualized by immunofluorescence (scale = 50 µm). (**A**–**D**) Figure adapted with permission from Calvier et al. Sci. Trans. Med., 2020 [53]. *n* ≥ 3; * *p* < 0.05 and ** *p* < 0.01.

**Figure 4 cells-13-00583-f004:**
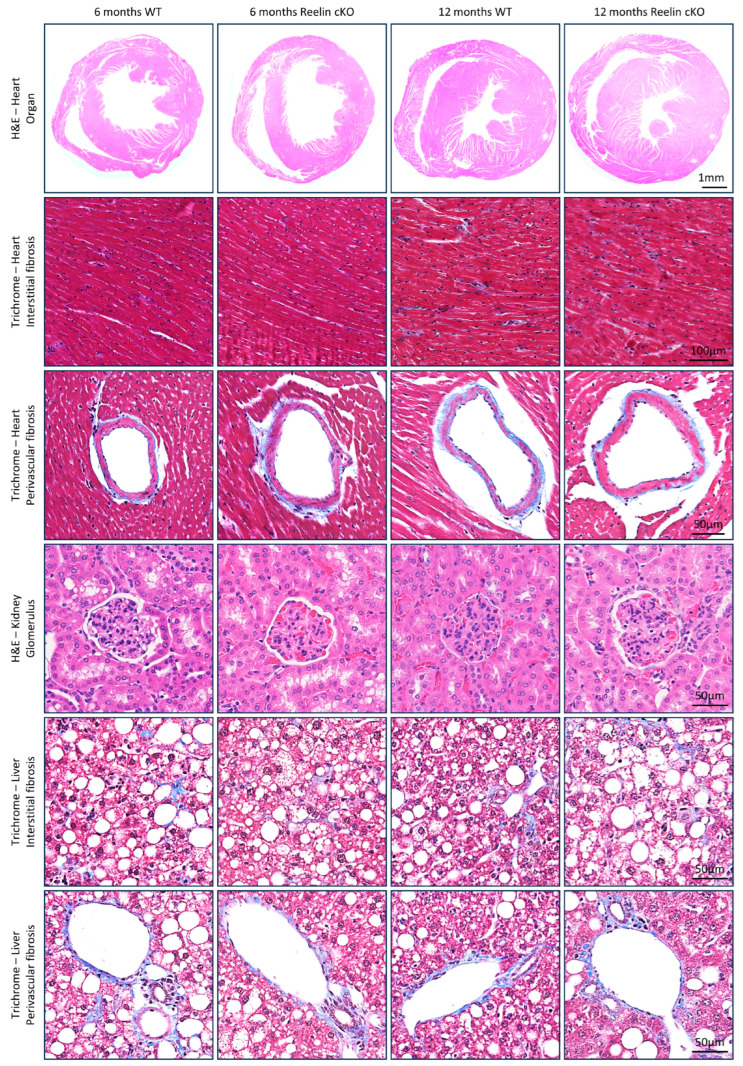
**Long-term Reelin depletion has no adverse effect on organ functions.** Reelin conditional KO mice and WT littermates, both on LDLR KO background, were fed a Western diet for 4 months (for the 6-month time point) or for 10 months (for the 12-month time point). At the end of each time point, various organs including the heart, kidney, and liver were analyzed to find any adverse effect of prolonged Reelin depletion under the physiological stress imposed by a Western diet. Representative pictures are shown using either H&E or Masson’s trichrome staining.

## Data Availability

Data are contained within the article.

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
