# Peer review of "Safety of Anti-Reelin Therapeutic Approaches for Chronic Inflammatory Diseases"

_cells, 2024, doi:10.3390/cells13070583_

Round 1

Reviewer 1 Report

Comments and Suggestions for Authors

The manuscript by Calvier et al. examines the safety and potential effects of depleting plasma Reelin through peripheral antibodies for chronic inflammatory diseases. The authors observed that Reelin depletion selectively modulates endothelial responses, reducing pathological leukocyte infiltration while maintaining basal diapedesis, and preserves its crucial functions in the central nervous system without adverse effects on synaptic plasticity. Additionally, the authors show that long-term depletion under metabolic stress has not revealed any adverse effects on the physiology of vital organs.

The manuscript is well-written, well-designed, and the results support the conclusions presented by the authors. The study presents very interesting and promising data. The authors demonstrate that the treatment does not decrease the expression of Reelin in the CNS, but they do not provide information about its expression in other tissues. If the treatment reduces the expression in other tissues, the authors should evaluate and discuss this aspect, at least in the discussion section of the manuscript. Other studies have shown that the upregulation of Reelin in response to initial injuries in chronic inflammatory processes might prevent the onset of cancer, as observed in the colon (https://doi.org/10.3390/biology11101406) and other tissues. The downregulation of Reelin within tissues could promote the development of cancer processes.

Perhaps, in the future, the authors could test their promising research with other chronic inflammatory processes and evaluate its effects on these processes and its implications in cancer development.

Minor comments:

1.      The citation style and references format do not comply with the journal's guidelines.

2.      The authors' list is missing under the title.

Author Response

See response attached

Reviewer 2 Report

Comments and Suggestions for Authors

The manuscript demonstrated the safety of anti-Reelin as a potential therapy for chronic inflammation diseases. The data are reliable, confirming from multiple ways that downregulation of Reelin is a safe anti-inflammatory therapy and have clinical significance. I have only the following minor concerns:

1. The legend in the Figure 2 is confusing. For example, the mention of "One total brain hemisphere was lysed" in Line 220-221 should refer to Figure 2B instead of 2A. Similarly, the reference to "spinal cord" in Line 224 should pertain to Fig 2C, etc. Please recheck this paragraph.

2. The conditional knock out mice used in Fig 4 do not indicate the time point of Tamoxifen-induced knockout in the Methods section. Additionally, there is no evidence provided regarding the efficiency of Reelin downregulation in various organs or plasma. Moreover, a statistical graph should be added to demonstrate the lack of damage to cardiac function, including the thickness of the cardiac ventricular wall.

Author Response

See response attached
